# SPF: A spatial and functional data analytic approach to cell imaging data

Thao Vu[1], Julia Wrobel[1], Benjamin G. Bitler[2,3], Erin L. Schenk[4], Kimberly R. Jordan[5], Debashis Ghosh[1]*

1 Department of Biostatistics and Informatics, University of Colorado Anschutz Medical Campus, Aurora, Colorado, United States of America, 2 University of Colorado Comprehensive Cancer Center, Aurora, Colorado, United States of America, 3 Department of OB/GYN, Division of Reproductive Sciences, The University of Colorado, Aurora, Colorado, United States of America, 4 Department of Medicine, University of Colorado Anschutz Medical Campus, Aurora, Colorado, United States of America, 5 Department of Immunology and Microbiology, University of Colorado School of Medicine, Aurora, Colorado, United States of America

* debashis.ghosh@cuanschutz.edu

## Abstract

The tumor microenvironment (TME), which characterizes the tumor and its surroundings, plays a critical role in understanding cancer development and progression. Recent advances in imaging techniques enable researchers to study spatial structure of the TME at a single-cell level. Investigating spatial patterns and interactions of cell subtypes within the TME provides useful insights into how cells with different biological purposes behave, which may consequentially impact a subject's clinical outcomes. We utilize a class of well-known spatial summary statistics, the K-function and its variants, to explore inter-cell dependence as a function of distances between cells. Using techniques from functional data analysis, we introduce an approach to model the association between these summary spatial functions and subject-level outcomes, while controlling for other clinical scalar predictors such as age and disease stage. In particular, we leverage the additive functional Cox regression model (AFCM) to study the nonlinear impact of spatial interaction between tumor and stromal cells on overall survival in patients with non-small cell lung cancer, using multiplex immunohistochemistry (mIHC) data. The applicability of our approach is further validated using a publicly available multiplexed ion beam imaging (MIBI) triple-negative breast cancer dataset.

## Author summary

Investigating spatial patterns and interactions of cells in the tumor microenvironment (TME) provides useful insights into cancer development and progression. In this work, we proposed a novel approach which combined established spatial summary functions with functional data analysis to flexibly model the cell-cell interactions with overall survival at different inter-cell distances, in conjunction with other clinical predictors such as age, disease stage. By applying the proposed framework to multiplex immunohistochemistry (mIHC) data of patients with non-small cell lung cancer (NSCLC), we studied the nonlinear impact of spatial interactions between tumor and stromal cells on overall survival.

**Data Availability Statement:** The TNBC dataset is publicly available at https://www.angelolab.com/mibi-data. The mIHC datasets (NSCLC and ovarian cancer) are part of the VectraPolarisData package, available at https://github.com/julia-wrobel/

VectraPolarisData. Open-source code and reproducible analysis script can be accessed at https://github.com/thaovu1/SPF.

**Funding:** B.G.B. is supported by the Department of Defense Award (OC170228) and an American Cancer Society Research Scholar Award (134106-RSG-19-129-01-DDC). E.L.S. is supported by NIH grant K12 CA086913 and ACS IRG #16-184-56 from the American Cancer Society to the University of Colorado Cancer Center, and a grant from the Cancer League of Colorado. The funders had no role in study design, data collection and analysis, decision to publish, or preparation of the manuscript.

**Competing interests:** The authors have declared that no competing interests exist.

The applicability of our proposed method is further validated using a publicly available multiplexed ion beam imaging (MIBI) triple-negative breast cancer (TNBC) dataset.

This is a *PLOS Computational Biology* Methods paper.

# 1 Introduction

The tumor microenvironment (TME), which consists of tumor cells, stromal cells, immune cells and the extracellular matrix, plays a critical role in understanding cancer development and progression [1, 2]. The concept of the TME has been around for several centuries; it was first documented by Virchow in 1863, characterizing the relationship between inflammation and tumor pathology [3, 4]. Later, the emergence of Paget's "seed and soil" principle in 1889 further emphasized the relationship between primary tumors and their microenvironment in influencing tumor evolution [5]. However, many TME studies were not widely recognized in the field of cancer research until the late 1970s and 1980s [4]. Then, there were a number of novel discoveries indicating the influences of TME-induced signals on cancer cells and their progression [6–9]. In particular, an experimental model proposed by Tarin et al. highlighted the role of microenvironments in metastasis potential of primary tumors in mice [10]. Additionally, in 1989, Ferrara's research group discovered vascular endothelial growth factor (VEGF) and its ability to induce angiogenesis, which then became a driving force for anti-angiogenic cancer research [11].

The TME is known to be complex and heterogeneous due to the continuous cellular and molecular adaptations in primary tumor cells that allow for tumor growth and proliferation. Accurately characterizing such heterogeneity is essential in gaining a better understanding of cancer and developing more effective treatment strategies. With advancements in technology, great progress has been made in high parameter imaging of tissues in situ to allow simultaneous quantification and visualization of individual cells in tissue sections. More specifically, multiplex tissue imaging (MTI) [12] methods such as cyclic immunoflourescence (CyCIF) [13], CO-Dectection by indEXing (CODEX) [14], multiplex immunohistochemistry (mIHC) [15], imaging mass cytometry (IMC) [16], and multiplex ion beam imaging (MIBI) [17] are capable of measuring the expression of tens of markers at single-cell resolution while preserving the spatial distribution of cells. As an example, multiplex immunohistochemistry (mIHC) detects and visualizes specific antigens in cells of a tissue section by utilizing antibody-antigen reactions coupled to a flourescent dye or an enzyme [18, 19]. Another instance includes multiplexed ion beam imaging (MIBI) [17], which utilizes secondary ion mass spectrometry to image metal-conjugated antibodies. As such, MIBI enables single-cell analysis of up to 100 parameters without spectral overlap between channels. Altogether, imaging provides an additional dimension of spatial resolution to the single cell signature profiles, which in turn allows researchers not only to study cellular composition but also to make inferences about specific cell-cell interactions.

As individual cells within the TME are genetically and epigenetically varied, they are competing with each other for space and resources (i.e., oxygen, nutrients, etc.) [20]. This is analogous to diverse species in their natural habitats as seen in ecology [21]. Typical ecological studies often involves examining spatial structures of species in a given habitat. Thus, leveraging analysis tools developed in ecology may be beneficial in studying spatial cell-cell interactions in the tumor ecosystem. For instance, Alfarouk et al. [22] highlighted regional variations in the distribution of cancer cells in relative proximity to blood vessels. This has an immediate

analogy to vegetation around waterways in a riparian ecosystem. Statistics characterizing the distribution of spatial distances are typically used to investigate spatial patterns of cells in the TME [23]. For example, some quantitative measures exist, such as the Morisita-Horn index [24], which has been employed by Maley et al. [25] to capture similarity between two local communities, i.e., tumor and immune cells in breast cancer. In addition, the intratumor lymphocyte ratio is another measure used to quantify the degree of infiltration of immune cells into the tumor [26], which has been shown to have prognostic potential.

In spatial statistics terminology, a collection of cells in the TME can be considered as a point pattern generated from a point process. If the cells are a realization of a spatial process assuming complete spatial randomness (CSR), then cells do not have preference for any spatial location. In other words, cells are randomly scattered in a given region of study, in this case the TME. Under this assumption, any deviations in spatial patterns from the null model of CSR could potentially provide some useful insights into how genetically heterogeneous cells behave. In addition to spatial locations, each point in a pattern can be associated with attributes as referred to as a "mark", which can be numerical (e.g., expression intensity for a given protein, cell size, etc.) and/or categorical (e.g., cell types: immune cells, tumor cells, macrophages, etc.). Such a point pattern is known as a marked point process. The K-function, a popular summary statistic proposed by Ripley [23], has been used to capture interpoint dependence with regard to distances between points in a point pattern. Several transformations of the K-function have also been introduced to explore not only spatial association of points but also the variation in the corresponding mark values. Patrick et al. recently utilizes the K-function as an exploratory analysis tool to identify and summarize complex spatial localization of multiple cell types within the TME, with applications on multiplexed imaging cytometry data [27]. Similarly, Canete et al. introduced the R package `spicyR` to relate changes in spatial localization of different cell types across subjects with disease progression [28], utilizing a localization score. The score, which is calculated as a integrated difference between the observed and expected L-function, i.e., the variance stabilized K-function, summarizes the spatial attraction or avoidance between pairs of different cell types. Following a different approach, Barua et al. considers the G-cross function to quantitatively differentiate the colocalization between tumor cells and infiltrative immune cells versus tumor cells and noninfiltrative cells as a function of cell distance [29]. Area under G-cross curves (AUC) is then incorporated with clinical factors including patient age, smoking history, and disease stage to find association with patient overall survival through univariate Cox proportional hazard regression model [30]. However, summarizing each curve into a single value might lose some information regarding the progression patterns as the inter-cell distance increases.

Herein, we introduce an approach, which leverages both spatial statistic summaries as well as a recently published model, the Additive Functional Cox Model (AFCM) [31], to incorporate spatial heterogeneity of cells available in histological images as functional covariates in a regression framework. Such an approach allows for the addition of other clinical variables to study their impact on patient survival. More precisely, we employ a derivative of the K-function, the m̲ark c̲onnection f̲unction (mcf) [32], to qualitatively investigate the spatial patterns of the "seed" and "soil" factors in the TME such as primary tumor versus stromal cells, primary tumor versus immune cells, etc. In addition, the correlation in the expression level of each marker between neighboring cell types can be quantified using Moran's I statistic [33]. Depending on the research problem, the proposed framework provides the flexibility in modelling qualitative and/or quantitative spatial information. The remainder of the article is organized as follows. Section 2 describes in detail summary spatial functions and how to incorporate them into the Cox model. Section 3 presents the results on real datasets, and Section 4

describes the simulation framework. Finally, we provide concluding remarks and discussion in Section 5.

## 2 Materials and methods

The main goal of our proposed approach is to model the spatial heterogeneity of cells in histological images and patient overall survival in a functional data analysis framework, in addition to other scalar clinical variables. As such, summary functions capturing the spatial architecture of cells in each image are necessary inputs of our model. We utilize mark connection function (mcf) [32] and Moran's I statistic [33] to represent qualitative and quantitative spatial information, respectively. We then model the available spatial input functions using the Additive Functional Cox Model. Details are given below.

### 2.1 Spatial summary functions

**Summary functions for qualitatively marked point patterns.** Qualitatively marked patterns consist of different types of points. An example includes cells in a given image with labels showing different cell types or tissue categories. Some numerical measures are typically employed to summarize any important feature of a given point pattern such as nearest-neighbor distance, which measures the average distance from a cell of one type to its closest cell of other type in the same pattern. Furthermore, some spatial functions are also used to further detect and quantify patterns using the density of points that are $r$ units apart. In particular, the mark connection function (mcf) [32, 34] is used to reveal interesting relationships between points belonging to two different types $i$ and $j$, referred to as **cross-type points**.

Let $g_{ij}(r)$ be the pair correlation function between types $i$ and $j$ and $g(r)$ be the pair correlation function for an unmarked process which disregards any label associated with points in the pattern. This pair correlation function $g(r)$ is used as an alternative to the K-function and describes the distribution of interpoint distances equal to $r$ and, for the unmarked process, is defined by

$$g(r) = K'(r)/2\pi r,$$

where $K'(r)$ is the derivative of the K-function with respect to distance $r$. Similarly,

$$g_{ij}(r) = K'_{ij}(r)/2\pi r$$

with $K'_{ij}(r)$ being the derivative of $K_{ij}(r)$. Note that $K'_{ij}(r)$ is the cross-type K-function capturing the expected number of points of type $j$ lying within $r$ unit distance of a typical point of type $i$.

Then, we define the mcf as

$$mcf_{ij}(r) = \frac{P(\text{point of type } i \text{ in U, point of type } j \text{ in V})}{P(\text{point in U, point in V})} = \frac{\lambda_i \lambda_j g_{ij}(r)}{\lambda_\bullet g(r)}, \tag{1}$$

where $\lambda_i$ and $\lambda_j$ are intensity functions of types $i$ and $j$ respectively; $\lambda_\bullet$ is the intensity function of unmarked point process. U and V are two separate subsets of the pattern, referred to as **subpatterns**, which contain points of types $i$ and $j$, respectively. Here, $\lambda_i$ and $\lambda_j$ are empirically estimated as the ratio of the number of points of U and V and the image area W. In other words, $\lambda_i = n(u)/|W|$ and $\lambda_j = n(v)/|W|$ with $n(u)$ and $n(v)$ the number of observed points in $u$ and $v$ respectively, and $|W|$ as the area of an observed image. Similarly, the unmarked intensity is estimated as the sum of individual marked intensity functions in a point process, as $\lambda_\bullet = \lambda_i + \lambda_j$. Higher values of the mark-connection function denote more cooccurrences of points of types $i$ and $j$, while the reverse holds for smaller values. Also note that the mark connection

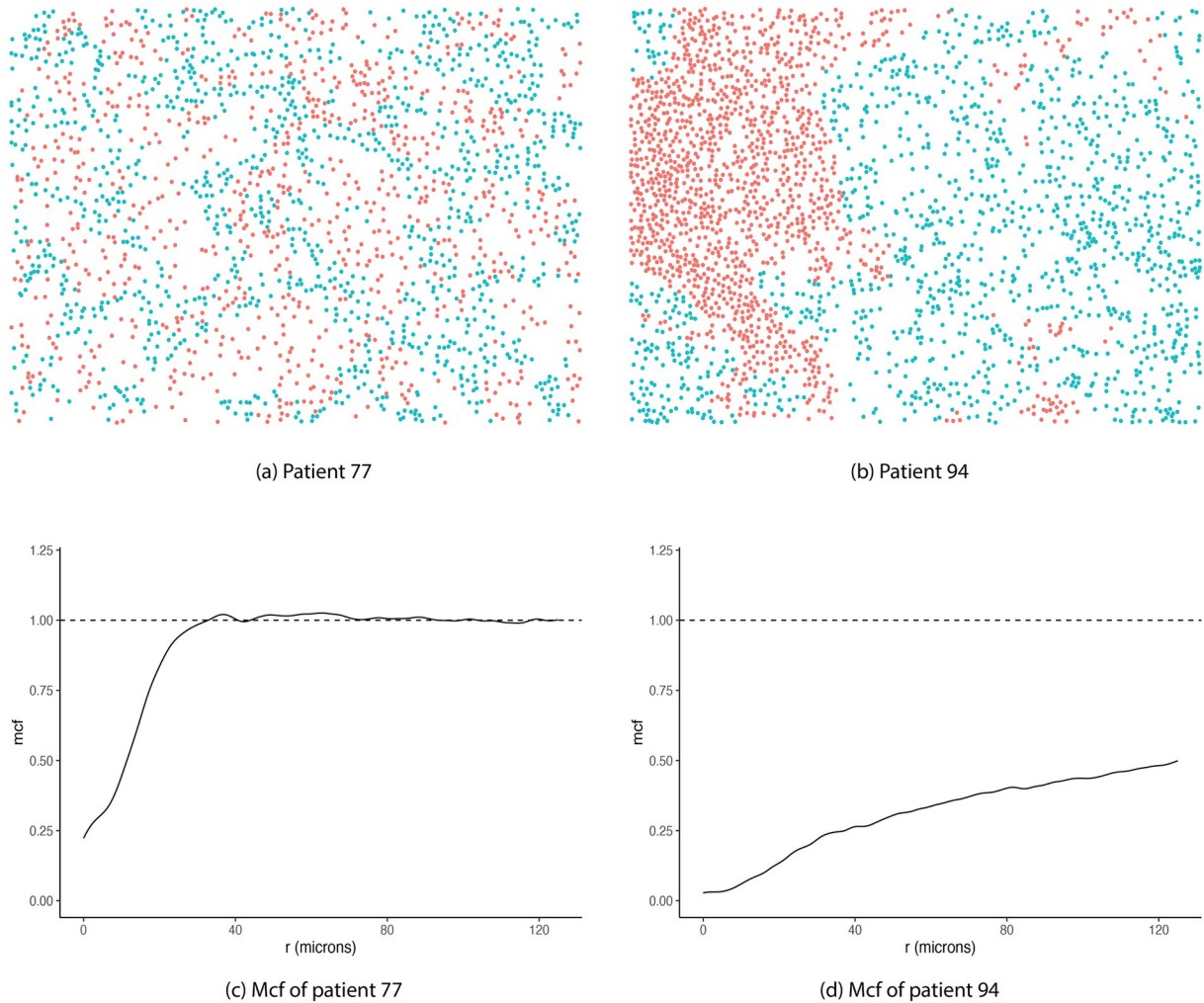

**Fig 1. Examples of mark connection functions (mcf).** Tumor (turquoise) and stromal (red) cell distributions of two patients' images: (A) 77 and (B) 94, respectively. Corresponding mcf curves are shown in (C) and (D), respectively. Dashed horizontal lines representing mcf under complete randomness. Mcf values below 1 indicate strong clustering of cells of same type, while values above 1 suggest higher level of mixing in of the two cell types.

makes an assumption of isotropy, i.e., the function value between two points only depends on the distance between the two points and not on the location.

Panels (A) and (B) of Fig 1 illustrate the distributions of tumor and stromal cells in two representative subjects, while Fig 1C and 1D display the mcf curves that capture the spatial interactions of the tumor and stroma subpatterns with respect to inter-cell distance. At short distances (small $r$ values), cells of each type tend to cluster, leading to mcf values below the complete randomness indicating line at 1. However, as $r$ increases, stromal and tumor cells start mixing in (Fig 1A), which leads to more cross-type cell interactions. As a result, the corresponding mcf values become slightly larger than 1 toward the end of the curve (Fig 1C).

**Summary functions for numerical marks of cross-type pairs of points.** While the aforementioned mcf provides a measure of spatial proximity of cross-type points, Moran's I correlation [33] incorporates additional information by taking into account a continuous value associated with each subpattern. For example, if the two subpatterns of interest are tumor and

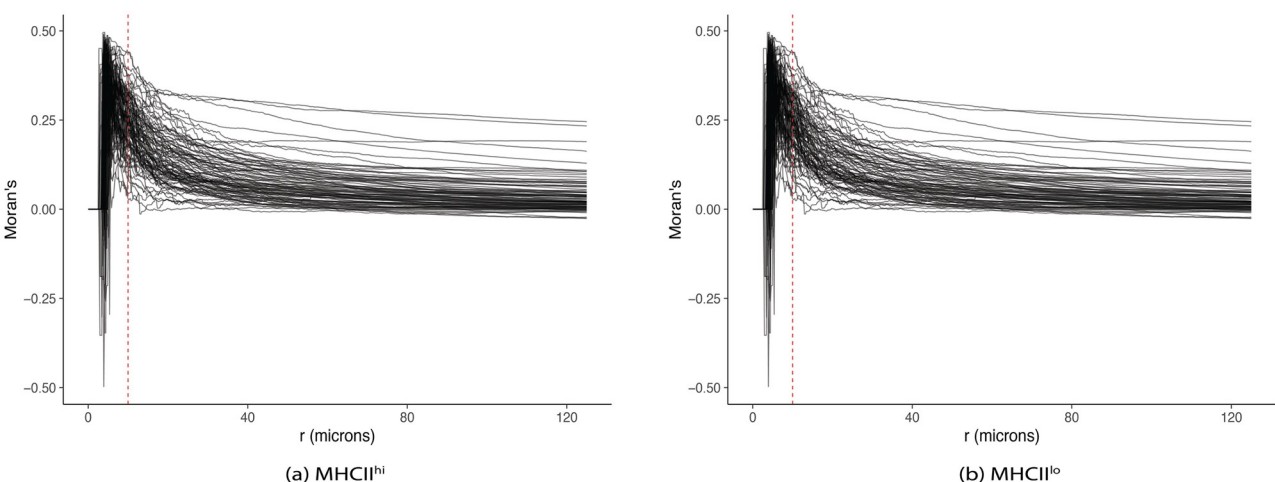

**Fig 2. Moran's I correlation between tumor and stromal cells using MHCII in NSCLC.** (A) MHCII$^{hi}$ group. (B) MHCII$^{lo}$ group. Patients with more than 5% tumor cells positively expressed with MHCII were classified into MHCII$^{hi}$ group. Negative Moran's I values indicate an inverse relationship in MHCII expression between tumor and stromal cells. Moran's I values above 0 suggest a direct relationship in MHCII expression between tumor and stromal cells. Dashed red vertical line in each panel corresponds to distance $r = 10 \ \mu$m.

stromal cells, Moran's I allows quantification of subpattern interaction plus additional incorporation of a protein functional marker such as MHCII. The Moran's I correlation of the $m$th mark between points of the two subpatterns $i$ and $j$, which are $r$ units apart, denoted as $I_m^{ij}(r)$ is calculated as follows

$$I_m^{ij}(r) = \frac{\sum_{s \in i}\sum_{t \in j}(m_s^i - \overline{m^i})(m_t^j - \overline{m^j})}{\sum_s (m_s^i - \overline{m^{pool}})^2 + \sum_t (m_t^j - \overline{m^{pool}})^2}, \tag{2}$$

where $m_s^i$ is the $s$th element of the $m$th mark in $i$ subpattern, $m_t^j$ is the $t$th element of the $m$th mark in $j$ subpattern. Note that $\overline{m^i}$ and $\overline{m^i}$ are the mean marker intensities of type $i$ and $j$ points, respectively, while $\overline{m^{pool}}$ is the overall mean intensity of both types. Fig 2 overlays Moran's I curves across all subjects, which depicts correlation in MHCII marker intensity between tumor and stromal cells as a function of distance $r$ for two groups of patients in the lung cancer dataset, (A) MHCII$^{hi}$ and (B) MHCII$^{lo}$ [35]. Similar patterns can be observed across the two panels. Specifically, tumor cells and their immediate neighboring stromal cells are negatively correlated in MHCII expression when they are less than 10 $\mu$m apart. As the cross-type cell distance increases, the correlation rises at different rates, and eventually drops close to 0. A more detailed discussion is included in Section 3.1.

## 2.2 Model

The aforementioned functions can be considered as summarized features capturing spatial interaction between individual cells for each subject. In order to investigate the association between cell-level spatial effect and patient survival, we leverage a previously published model, the Additive Functional Cox Model (AFCM) [31]. This model allows us to incorporate each subject's spatial summary function as functional covariates in addition to other scalar clinical variables such as age, sex, and disease stage. In particular, we model the log hazard function for each $i$th subject $i = 1, \ldots, N$ with $T_i$ and $C_i$ respectively denoting the corresponding survival time and censoring time, which are assumed to be conditionally independent given covariates

in the model. Note that due to right-censoring, we only observe $Y_i = \min(T_i, C_i)$. For $i = 1, \ldots, n$, let $\delta_i = I(T_i \leq C_i)$ serve as a censoring indicator. For each $i$th subject, we observe $p$ scalar clinical variables as $\mathbf{Z}_i = \{Z_{i1}, Z_{i2}, \ldots, Z_{ip}\}$ and functional covariates $\mathbf{X_i} = \{X_i(s)\}_{s \in \mathcal{S}}$ which is a continuous and integrable curve defined on a compact interval. We assume that $\mathbf{X}$ is observed on a grid of points. In our setting, $\mathbf{X}_i$ will be a spatial summary for subject $i$, such as a mark connection function or Moran's I function. Following Cui et al. [31], we jointly model the effect of cell-level spatial interactions with clinical variables on a subject's risk of mortality as follows

$$\log \lambda_i[t; \mathbf{Z}_i, \mathbf{X}_i] = \log \lambda_0(t) + \mathbf{Z}_i^T \boldsymbol{\beta} + \int_{\mathcal{S}} F\{s, X_i(s)\} ds, \tag{3}$$

where $\log \lambda_i[t; \mathbf{Z}_i, \mathbf{X}_i]$ is the log hazard at time $t$, given scalar covariates $\mathbf{Z}_i$ and functional covariates $\mathbf{X}(s)_i$, $i = 1, \ldots, n$. In Eq (3), $\log \lambda_0(t)$ is the log baseline hazard function, and the parameter vector $\boldsymbol{\beta}$ represents a multiplicative change in log hazard ratios for a one-unit increase in $\mathbf{Z}_i$. We also have $F$, an unspecified smooth function to be estimated. As described in McLean et al. [36], $F(.)$ can be modelled as a tensor product of two penalized spline bases:

$$F(s, x) = \sum_{j=1}^{K_s} \sum_{k=1}^{K_x} \theta_{j,k} B_j(s) B_k(x), \tag{4}$$

where $B_j(s)$ and $B_k(x)$ are splines defined on functional domain $s$ and functional covariate domain $x$, respectively. $\theta_{j,k}$ for $j = 1, \ldots, K_s$; $k = 1, \ldots, K_x$ are spline coefficients. Following the same approach as Cui et al. [31], we apply cubic regression splines for both domains $s$ and $x$. Combining Eqs (3) and (4), the model can be rewritten as:

$$
\begin{aligned}
\log \lambda_i[t; \mathbf{Z}_i, \mathbf{X}_i] &= \log \lambda_0(t) + \mathbf{Z}_i^T \boldsymbol{\beta} + \int_{\mathcal{S}} F\{s, X_i(s)\} ds \\
&= \log \lambda_0(t) + \mathbf{Z}_i^T \boldsymbol{\beta} + \sum_{j=1}^{K_s} \sum_{k=1}^{K_x} \theta_{j,k} \int_{\mathcal{S}} B_j(s) B_k\{X_i(s)\} ds \\
&= \log \lambda_0(t) + \mathbf{Z}_i^T \boldsymbol{\beta} + \mathbf{V}_i^T \boldsymbol{\theta} \\
&= \log \lambda_0(t) + \mathbf{W}_i^T \boldsymbol{\gamma}.
\end{aligned}
\tag{5}
$$

where, $\boldsymbol{\gamma}^T = (\boldsymbol{\beta}^T, \boldsymbol{\theta}^T)$, with $\boldsymbol{\theta}$ is a vector of entries $\theta_{j,k}$; $\mathbf{W}_i^T = (\mathbf{Z}_i^T, \mathbf{V}_i^T)$, with $\mathbf{V}_i^T$ is a vector of entries $\int_{\mathcal{S}} B_j(s) B_k\{X_i(s)\} ds$, $j = 1, 2, \ldots, K_s$, $k = 1, 2, \ldots, K_x$. The parameters $\boldsymbol{\gamma}$ are estimated by maximizing the penalized partial log-likelihood with the smoothing parameter imposed on $\boldsymbol{\beta}$. The penalized partial log-likelihood is defined as follows

$$
\begin{aligned}
l_p(\boldsymbol{\gamma}|\lambda) &= l(\boldsymbol{\gamma}) - \lambda J(\boldsymbol{\theta}) \\
&= \sum_{i=1}^N \delta_i \left\{ \mathbf{W}_i^T \boldsymbol{\gamma} - \log \sum_{Y_j \geq Y_i} \exp(\mathbf{W}_i^T \boldsymbol{\gamma}) \right\} - \lambda J(\boldsymbol{\theta})
\end{aligned}
\tag{6}
$$

where the penalty term can be expressed as a quadratic term $\lambda J(\boldsymbol{\theta}) = \frac{1}{2} \lambda \boldsymbol{\gamma}^T \boldsymbol{D} \boldsymbol{\gamma}$ with $\boldsymbol{D}$ is a symmetric, non-negative definite penalty matrix. For a given smoothing parameter $\lambda$, the regression coefficients are estimated as:

$$\hat{\boldsymbol{\gamma}}(\lambda) = \operatorname*{argmin}_{\boldsymbol{\gamma}} - l_p(\boldsymbol{\gamma}|\lambda)$$

using Newton-Raphson procedure, provided the gradient vector $\mathcal{G} = \partial l/\partial \boldsymbol{\gamma}$ and Hessian matrix $\mathcal{H} = \partial l^2/\partial \boldsymbol{\gamma} \partial \boldsymbol{\gamma}^T$. Following Wood et al. [37], the selection of the smoothing parameter λ is done by optimizing the log Laplace approximate marginal likelihood, which can be expressed as

$$\mathcal{V}(\lambda) = l_p(\boldsymbol{\gamma}) + \log |\boldsymbol{D}^\lambda|_+ - \frac{1}{2}\log |\mathcal{H}| + \frac{M_p}{2}\log(2\pi)$$

where, $\boldsymbol{D}^\lambda = \lambda \boldsymbol{D}$ and $|\boldsymbol{D}^\lambda|_+$ is the product of positive eigenvalues of $\boldsymbol{D}^\lambda$. $M_p$ is the number of zero eigenvalues of $\boldsymbol{D}^\lambda$. The process involves optimizing $\mathcal{V}$ with respect to log λ. More details can be found in [37].

### 2.3 Datasets

We used non-small cell lung cancer (NSCLC) and triple-negative breast cancer (TNBC) datasets collected using multiplex immunohistochemistry (mIHC) and multiplexed ion beam imaging (MIBI) platforms, respectively, to evaluate the applicability of our proposed model.

**Non-small cell lung cancer (NSCLC).**   Tissue slides collected from 153 patients with non-small cell lung cancer were sequentially stained with antibodies specific for CD19, CD8, CD3, CD14, major histocompatibility complex II (MHCII), cytokeratin, and DAPI; then the slides were imaged on the Vectra 3.0 microscope (Akoya Biosystems). The acquired images were then processed using Akoya's inForm tissue analysis software to obtain a data matrix with rows corresponding to individual cells and columns corresponding to x- and y-coordinates of each cell on the image, individual marker expression, and cell phenotypes. Each individual had three to five images corresponding to small regions of the tissue sample. Due to the sparsity issue of some images, we decided to select the image with the maximum number of cells to represent each subject. More details can be found in Johnson et al. [35].

**Triple-negative breast cancer (TNBC).**   TNBC biopsies were compiled into a tissue microarray (TMA) slides and stained with 36 antibodies targeting regulators of immune activation such as PD1, PD-L1, etc. The slides were imaged using the multiplexed ion beam imaging (MIBI) mass spectrometer. Cell segmentation was performed by adapting a convolutional neural network (CNN) approach, i.e., DeepCell [38] for nuclear segmentation to MIBI data. Details of the method were described in Keren et al. [39]. Images from 41 patients were processed using the R package raster to extract pixel coordinates. Panels (A) and (B) of S2 Fig show two representative pixel-level images for two randomly chosen patients. Since each cell consists of multiple pixels, the average pixel-level x- and y-coordinates were used to represent cell locations on each image. Panels (C) and (D) of S2 Fig illustrate cell-level images for patients 1 and 2, respectively, with dot size proportional to cell size and color-coded for each cell type (e.g., immune, endothelial, mesenchymal-like, tumor, etc.). With the cell information in the tumor and stroma regions of the TME not available, we focused on capturing the distribution of tumor and immune cells across images using mcf curves in this dataset. Additionally, two patients in the cohort did not have clinical information available regarding survival outcomes; and one patient's imaging data was corrupted with a high level of noise. As a result, only data of 38 patients were included in the model.

## 3 Application results

### 3.1 NSCLC

Fig 3A illustrates the locations of stromal and tumor cells in XY-coordinates for a representative image. A summary mcf using Eq (1) was calculated to characterize the spatial relationship between primary tumor cells and tumor-associated stromal cells as a function of distance

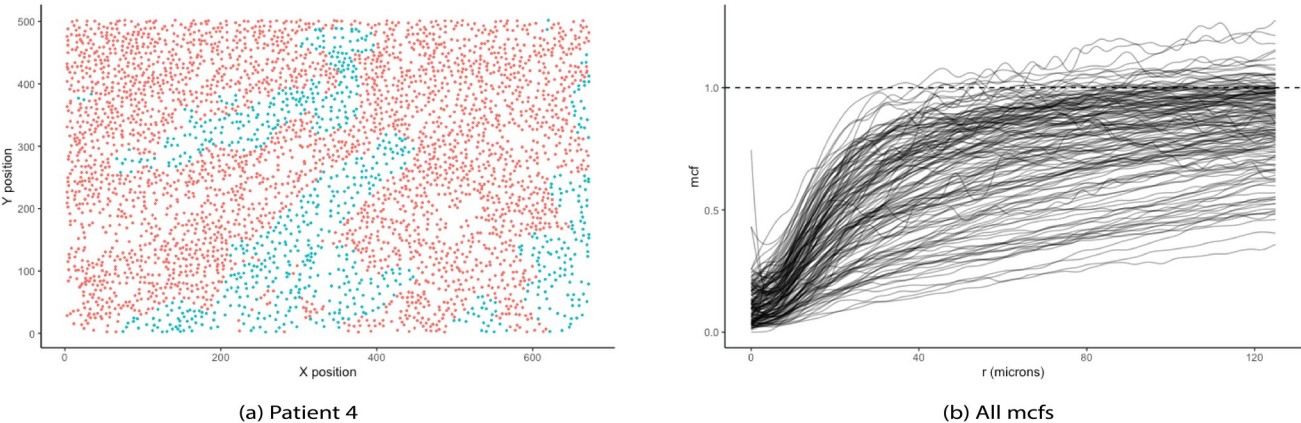

(a) Patient 4                                    (b) All mcfs

**Fig 3. NSCLC dataset.** (A) Representative image illustrating locations of stromal (red) and tumor cells (turquoise). (B) Mark connection functions (mcf) for the tumor—stromal cell distributions for all subjects. Dashed horizontal line represents mcf under complete randomness. Mcf values below 1 indicate strong clustering of cells of same type, while values above 1 suggest higher level of mixing in of the two cell types.

between cells. Since cell distance $r$ depended on the density of observed cells in a given image, we chose the image with the maximum number of cells to compute a reference range of distances. Fig 3B shows mcf curves for all subjects as a function of the distance between cells, measured in microns ($\mu$m). These mcf curves were used as functional covariates along with scalar clinical predictors such as total number of cells, disease stage, and age in model (5) to obtain the estimated hazard of mortality for each subject. Fig 4A highlights 30 mcf curves corresponding to 15 patients with shortest survival time (red) and 15 patients with longest observed censored time (black) to aid the interpretation of the fitted functional model. Fig 4C depicts the estimated functional surfaces $\hat{F}$ with values decreased from red to blue, corresponding to a decrease in hazard of mortality, while holding the remaining scalar variables constant. In particular, if there was strong clusterings of tumor cells in the neighborhood of 25 —75 $\mu m$, the overall survival improved, based on the negative estimate of the log hazard ratio. However, at distances beyond 75 $\mu m$, the more tumor cells clustered, the higher the risk of mortality, while the increased infiltration level of tumor-associated stromal cells reduced the hazard of death. Even though the mcf curves were calculated for cells in tumor and stroma tissue regions, more than 98% of cells in stroma region were immune cells including CD4$^+$, CD14$^+$, CD19$^+$, and CD8$^+$ T cells. Interestingly, this was in agreement with the conclusion from Johnson et al. that the increased spatial proximity of cancer cells to immune cells linked to better subject survival [35]. Section S.1.1 of S1 Text reported the summary output of the corresponding model. The p-value of 0.44 indicated no significant nonlinear association between the spatial interactions of tumor and stromal cells with overall survival. However, note that given a relatively large number of parameters to be estimated, our sample size might not provide enough power to detect such association. As a result, the qualitative interpretation of the model using the surface plots is still worthwhile.

Johnson et al. dichotomized lung cancer samples into high and low MHCII. In particular, specimens with more than 5% of lung cancer cells positive for MHCII were grouped into MHCII$^{hi}$ while the remaining samples were classified into MHCII$^{lo}$ group. The authors stated that the levels of immune infiltration increased in the MHCII$^{hi}$ TME, leading to a significantly improved overall survival. Motivated by this discovery, we explored the distributions of MHCII in cells from tumor and stroma regions separately. As illustrated in panel (A) of S1

Fig, we observed that in MHCII$^{hi}$ samples, the distribution of MHCII intensity across cells in stroma region was right-skewed while that marker expression of tumor cells were more symmetrically distributed and centered at a higher intensity. On the other hand, two distributions of MHCII in tumor and stromal cells had similar right-skewed shape for MHCII$^{lo}$ samples (panel (B) of S1 Fig). This inspired us to explore the correlation in MHCII between the two cell types as a function of cross-type cell distances via Moran's I metric according to Eq (2). Fig 2A and 2B illustrate the estimated Moran's I correlation curves for patients in MHCII$^{hi}$ and MHCII$^{lo}$ groups, respectively, as a function of distance between cross-type cells. There was slightly more variability between correlation curves of the subjects in the MHCII$^{hi}$ group as compared to that in MHCII$^{lo}$ group. The two groups shared a common trend of negative association in MHCII expression between tumor cells and their immediate neighboring stromal cells ($r \leq 10\ \mu m$). Note that in the 2D images captured by slicing 3D tissue samples, biological overlapping could potentially occur, however our 2D samples were sliced as thin as possible ($\approx 4\ \mu m$ per section) to minimize this potential overlap. Additionally, by investigating the diameters of tumor and stromal cells in the dataset, we noticed that the median diameter was about 7.2 $\mu$m. Altogether, it was possible to observe pairs of cross-type cells in the neighborhood less than 10 $\mu$m. Moreover, the tumor and stromal cells in such small radius span tended to have an inverse relationship in MHCII expression, i.e., tumor cells with low MHCII were likely to be adjacent to high MHCII stromal cells, leading to negative Moran's I correlation (Fig 2). As the neighborhood radius expanded, the correlation increased at various rates. Instead of categorizing subjects into MHCII$^{hi}$ and MHCII$^{lo}$ groups before fitting the Cox proportional model, we explored the direct impact of correlation of MHCII expression across tumor and stromal cells as a function of cross-type cell distance on survival outcome. Specifically, each Moran's I correlation curve served as a functional covariate in model (5). Fig 4B highlights 30 correlation curves corresponding to 15 patients with shortest survival times (red) and 15 patients with longest observed censored times (black) to aid the interpretation of the fitted functional model. Fig 4D depicts the estimated surface $\hat{F}$ with values decreased from red to blue, in correspondent with a decrease in hazard of mortality, holding the remaining scalar variables fixed. If stromal and tumor cells in the neighborhood radius between 25 and 75 $\mu$m were positively correlated in MHCII expression, the risk of mortality increased. However, as the positive relationship in MHCII continued past 100 $\mu$m, the estimated hazard of mortality decreased, linking to a better survival outcome.

### 3.2 TNBC

Keren et al. [39] classified subjects into "compartmentalized" and "mixed" using a mixing score, which was defined as a ratio of immune-tumor interactions to the total number of immune interactions. Fig 5A and 5B depict two representative images of the two categories. Taking a different approach, we utilized the mcf between tumor and immune cell distributions within each image to capture the tumor-immune interactions as a function of cell distance, measured in microns ($\mu m$). Fig 5C illustrates the progression of such interactions as the distance between cells increased, colored in black and red corresponding to "compartmentalized" and "mixed" categories, respectively. Particularly, when tumor and immune cells mixed in (e.g., Fig 5A), more interactions between tumor and immune cells were observed as the neighborhood radius expanded. As a result, the corresponding mcf curves increased at a faster rate, as compared to the ones associated with the compartmentalized cell distribution.

Due to the small sample size, we only included patient age as a scalar predictor, in addition to the functional covariates provided by mcf values (Fig 5C). The estimated functional surface $\hat{F}$ with values decreased from red to blue, corresponding to a drop from high to low hazard of

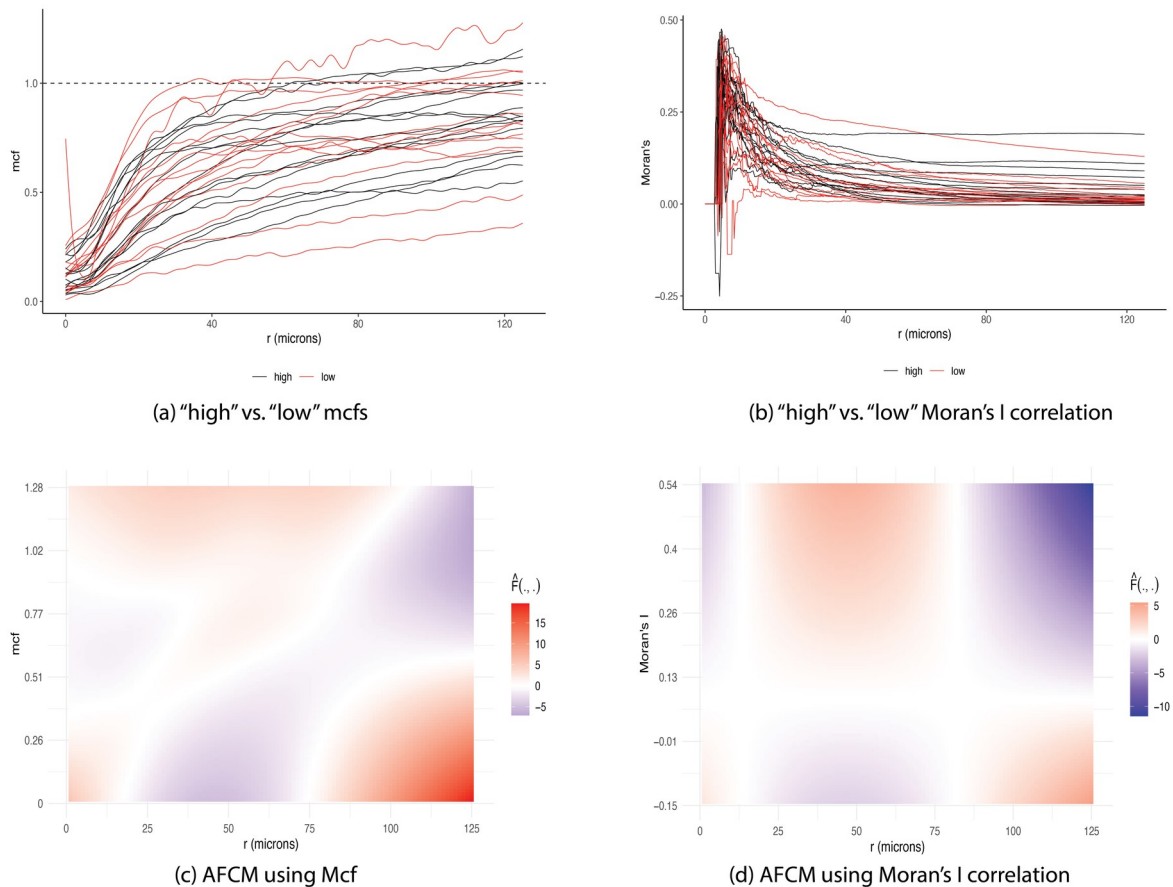

(a) "high" vs. "low" mcfs

(b) "high" vs. "low" Moran's I correlation

(c) AFCM using Mcf

(d) AFCM using Moran's I correlation

**Fig 4. Estimated surface from AFCM using NSCLC dataset.** (A) Mcf curves corresponding to "low" survival (i.e., shortest survival times) and "high" survival (i.e., longest observed censored times) patients in red and black, respectively. Note that mcf values below 1 indicate strong clustering of cells of same type, while values above 1 suggest higher level of mixing in of the two cell types. (B) Moran's I correlation using MHCII expression corresponding to "low" survival (i.e., shortest survival time) and "high" survival (i.e., longest observed censored time) patients in red and black, respectively. Moran's I values above 0 indicate a direct relationship in MHCII expression between tumor and stromal cells. Negative Moran's I values suggest an inverse association in MHCII expression between tumor and stromal cells. (C) Estimated surface from AFCM using mcf curves as functional covariates with values decreasing from positive (red) to negative (blue). (D) Estimated surface from AFCM using Moran's I correlation in MHCII as functional covariates with values decreased from positive (red) to negative (blue). Positive $\hat{F}$ corresponds to increased risk of mortality while negative $\hat{F}$ associates with reduced hazard of death.

mortality, while holding the scalar variable constant was shown in Fig 6C. In particular, if there were more immune cells surrounding tumor cells across all distances, which resulted in the mcf curves approaching the dashed horizontal line of 1 at a faster rate, the estimated log hazard ratio increased. On the contrary, when tumor and immune cells separated in their own compartments, corresponding to the mcf values less than 1, the hazard of death decreased.

Pan et al. [40] investigated the prognostic role of P53 in TNBC by applying Kaplan-Meier analyses on P53 positive and negative groups of patients. In particular, a sample was defined as P53 positive if any cancer cells was positively expressed. The authors concluded that P53 positivity associated with negative prognostic significance in breast cancer patients. Inspired by the findings, we investigated the distributions of P53 expression in tumor and immune cells separately for each patient. We observed that in P53 positive patients, the distribution of P53 intensity in immune cells was extremely right-skewed while that marker expression of tumor cells

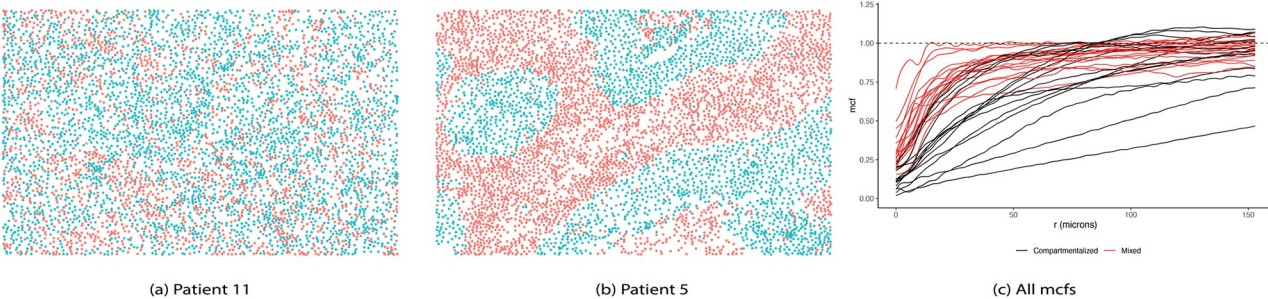

(a) Patient 11

(b) Patient 5

(c) All mcfs

**Fig 5. "Mixed" vs. "Compartmentalized" cell distributions in TNBC dataset.** (A) Cell-level image of patient 11 with "mixed" cell distribution. (B) Cell-level image of patient 5 with "compartmentalized" cell distribution. (C) Mark connection functions (mcf) in correspondence with "mixed" and "compartmentalized" tumor—immune cell distributions, colored by red and black, respectively. Dashed horizontal line represents mcf under complete randomness. Mcf values below 1 indicate strong clustering of cells of same type, while values above 1 suggest higher level of mixing in of the two cell types.

were distributed more symmetrically and centered at a much higher intensity, as shown in panel (A) of S3 Fig. On the other hand, the P53 expression distributions were right-skewed, similarly between tumor and immune cells in P53 negative patients (panel (B) of S3 Fig). In other words, tumor and immune cells in P53 negative samples were more likely to be correlated while it might not be the case for P53 positive samples. This motivated us to explore the correlation in P53 between the two cell types as a function of cross-type cell distance (Fig 6B) without necessarily dichotomizing samples into "P53 positive" vs "P53 negative" groups, via Moran's I metric (Eq 2). In a few patients, immune and tumor cells lying within 100 $\mu$m neighborhood radius, were positively correlated. As cell distance became larger, the correlation dropped close to 0. Though the correlation was rather weak, it was still worth studying the impact of the relationship between tumor and immune cells with respect to the P53 expression on patient survival using the proposed approach. In particular, the correlation curves served as functional covariates in the AFCM model in (5) in conjunction with age as a scalar predictor. The estimated functional surface $\hat{F}$ shown in Fig 6D had the values decreased from red to blue, in correspondence with a decrease in hazard of mortality, while holding the scalar variable fixed. Interestingly, positive correlation in P53 expression between immune and tumor cells across distances associated with a decline in estimated risk of mortality.

Section S.3 of S1 Text includes an additional analysis using a Vectra dataset of 114 ovarian cancer patients [41]. We followed the same approach to investigate the impact of spatial distribution of cells in tumor and stroma regions of the TME on patient overall survival. Furthermore, the correlation in Ki67 marker expression between tumor and stromal cells in close proximity was included in the model in (5) to study the prognostic value of Ki67 in ovarian cancer patients.

## 4 Simulation study

### 4.1 Setup

We performed simulation studies to evaluate the finite-sample properties of the proposed methodology. We considered a scenario with one functional covariate $\mathbf{X_i}$ and one scalar

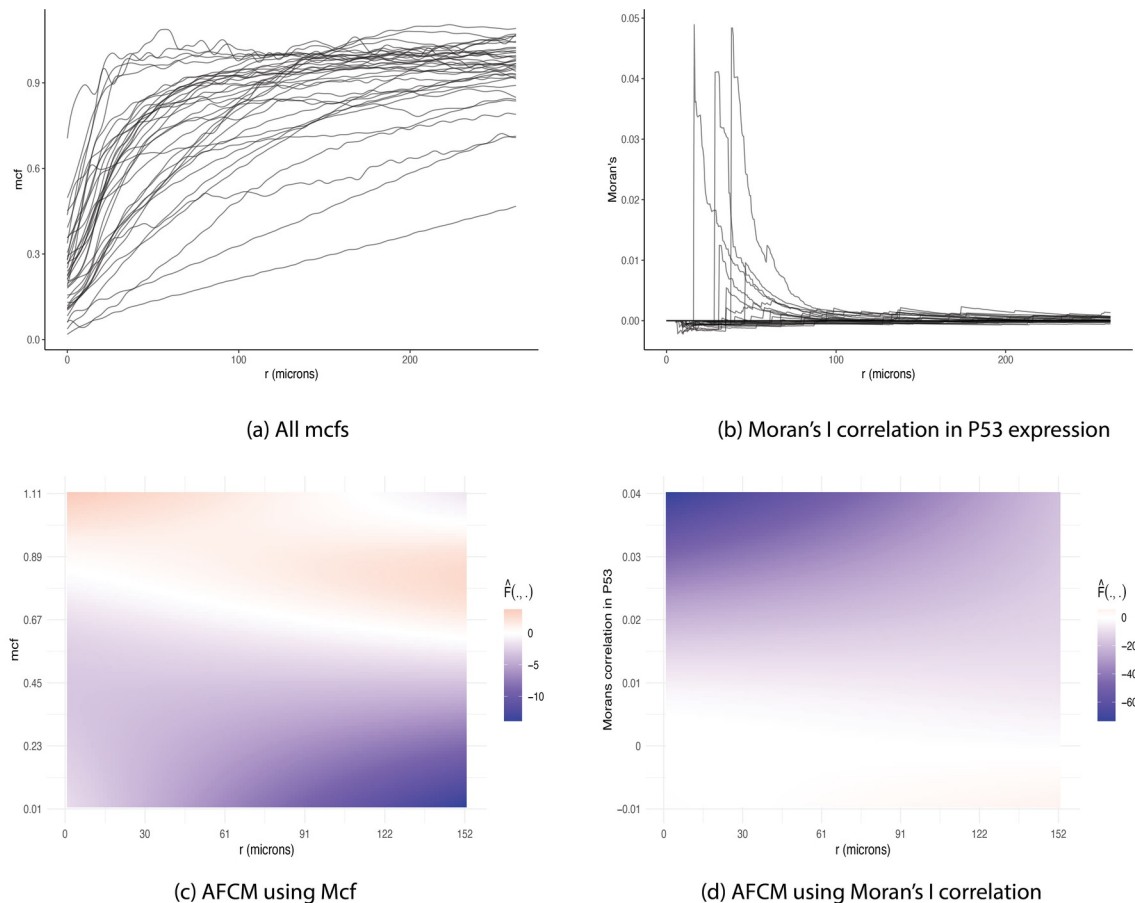

(a) All mcfs

(b) Moran's I correlation in P53 expression

(c) AFCM using Mcf

(d) AFCM using Moran's I correlation

**Fig 6. TNBC dataset.** (A) Mark connection functions (mcf) for the tumor—immune cell distributions for all subjects in TNBC dataset. Dashed horizontal line represents mcf under complete randomness. Mcf values below 1 indicate strong clustering of cells of same type, while values above 1 suggest higher level of mixing in of the two cell types. (B) Moran's I correlation between tumor and immune cells across subjects using P53 marker expression. Moran's I values above 0 indicate a direct relationship in P53 expression between tumor and immune cells. Negative Moran's I values suggest an inverse association in P53 expression between tumor and immune cells. (C) Estimated surface from AFCM using mcf curves as functional covariates, with values decreasing from positive (red) to negative (blue). Positive $\hat{F}$ corresponds to increased risk of mortality while negative $\hat{F}$ associates with reduced hazard of death. The more immune cells surrounding tumor cells across all distances (i.e., mcf values > 1), the higher risk of mortality (i.e., $\hat{F} > 0$). (D) Estimated surface from AFCM using Moran's correlation in P53 expression between tumor and immune cells, with values of $\hat{F}$ decreasing from positive (red) to negative (blue). Weak positive correlation in P53 expression between tumor and immune cells across all distances associates with a declined in risk of mortality with $\hat{F} < 0$.

covariate $Z_i$ for simplicity. Model (3) can be rewritten as:

$$\log \lambda_i(t; Z_i, \mathbf{X}_i) = \log \lambda_0(t) + Z_i\beta + \int_{\mathcal{S}} F\{s, X_i(s)\} ds$$

$$\log \lambda_i(t; Z_i, \mathbf{X}_i) = \log \lambda_0(t) + \eta_i$$

(7)

Both scalar and functional terms were simulated directly from the lung cancer dataset to mimic real-world parameter settings. More specifically, the scalar predictor $Z_i^*$ was simulated from the normal distribution with mean and standard deviation obtained empirically from the distribution of age. Additionally, the functional covariates $X_i(s)$ were simulated empirically by applying FPCA [42] to the estimated mcf curves. Following Cui et al. [31], we simulated the

functional covariates as $X_i^*(s) = \hat{\mu}(s) + \sum_{j=1}^{M} \sqrt{\hat{\lambda}_j} e_{ij} \hat{\phi}_j(s)$ with M being the number of principal components. In the definition of $X_i^*(s)$, $e_{ij}$ were random Gaussian noise with mean 0 and variance one. The mean function $\hat{\mu}(s)$, eigenvalues $\hat{\lambda}_j$, and eigenfunctions $\hat{\phi}_j(s)$ were computed by applying FPCA on the estimated mcf curves using the package `refund` in R [43]. By specifying the scalar coefficient $\beta = 1$ and the functional form $F(s, x) = x^3 s$, the simulated linear predictor was generated as $\eta_i^* = Z_i^* + \sum_{s \in \mathcal{S}} X_i^{*3}(s)s$. With the prespecified functional form $F$, we scaled the functional domain $\mathcal{S}$, which represented the distance between cells, to [0, 1]. The functional covariate values $\mathcal{X}$ were kept in the range [0, 1.2]. Doing so prevented the simulated linear predictor from being dominated by either term. From the fitted model in Section 3.1, we obtained the estimated cumulative baseline hazard $\int_0^t \lambda_0(x)dx$, which was then used to generate a survival function for each individual, such that $\tilde{S}_i(t) = \exp\{-e^{\eta_i^*} \int_0^t \lambda_0(x)dx\}$. The estimated survival times $T_i^*$ were generated from the survival function; and the censoring times $C_i^*$ were simulated based on the empirical distribution of the observed censoring times.

## 4.2 Predictive performance

Four datasets of different sizes (N = 1000, 500, 200, and 100, respectively) were simulated following the procedure in Section 4.1. Each dataset was partitioned into training (75%) and testing (25%) sets. Three models were fit using the training set: (1) AFCM model with both scalar and functional terms, (2) AFCM model with only functional term, and (3) regular Cox proportional hazard model with only scalar term. The predicted linear predictor $\hat{\eta}_i^{(u)}$, $u = 1, 2, 3$ was obtained from the testing set for the $u$th model. At each sample size, mean squared errors $MSE^{(u)}$ was computed as the average of squared differences between the predicted $\hat{\eta}_i^{(u)}$ and the "true" linear predictor $\eta_i^*$ such that $MSE^{(u)} = N_t^{-1} \sum_{i=1}^{N_t} (\hat{\eta}_i^{(u)} - \eta_i^*)^2$, with $N_t$ denoting the number of subjects in the testing set.

We repeated the simulation for 100 iterations and recorded the average MSE for each of the three models across four sample sizes N = 1000, 500, 200, 100 in Table 1. Fig 7 displays the distribution of MSEs for the three models at each sample size, respectively. As expected, the average MSEs and corresponding standard deviations increased as the sample size decreased from 1000 to 100. More precisely, as shown in Fig 7, the upper quartile MSE of the model (1) was less than the $25^{th}$ percentile of MSEs of the remaining two models (2) and (3), across all four sample sizes. The discrepancy between these quantiles became more apparent as the sample size increased. However, regardless of the sample size, model (1), which included both functional and scalar terms, performed substantially better than the other two models (2) and (3).

## 5 Conclusions and discussion

Interactions between cells within the TME play a crucial role in understanding cancer development and progression. With advances in imaging technology, the additional dimension of spatial resolution is achievable in addition to the single cell profiles. Conventional approaches summarize the spatial information between cells into a single quantitative value (e.g., Morisita-Horn index, intratumor lymphocyte ratio, etc.) before fitting a Cox proportional hazard model to quantify the association between cell profiles and survival outcome. Though easy to implement, these scalar summaries are unable to embed cell interactions as a function of spatial proximity in models that test associations with clinical outcomes. Alternatively, we consider cells with corresponding cell locations and features within the TME as marked point patterns. Then, the relative spatial organization of primary tumor cells and a variety of tumor-associated stromal cells as well as immune cells can be described through spatial summary

**Table 1. Mean squared errors (MSE) across three different models.** Different sample sizes are 1000, 500, 200, and 100, respectively. Corresponding standard deviations are recorded in parentheses.

| Model | N = 1000 | N = 500 | N = 200 | N = 100 |
|---|---|---|---|---|
| (1) Both | 0.64 (0.49) | 0.90 (1.09) | 0.73 (0.80) | 1.00 (1.13) |
| (2) Functional covariate | 1.86 (0.66) | 2.11 (1.21) | 1.88 (0.95) | 2.16 (1.48) |
| (3) Scalar covariate | 2.99 (1.29) | 3.28 (2.02) | 2.73 (1.88) | 2.79 (2.26) |

functions such as the mark connection function (for qualitative marks) or Moran's I correlation (for quantitative marks). We propose an approach to incorporate the resulting spatial functions as functional covariates into an additive function Cox model to investigate the impact of spatial structure of cells in the TME on overall survival in addition to some clinical predictors (e.g., age, disease stage, total cell counts, etc.). We demonstrate the applicability of the proposed method by analyzing multiplex imaging datasets from three separate cancer applications collected under two different (Vectra and MIBI) imaging platforms: NSCLC,

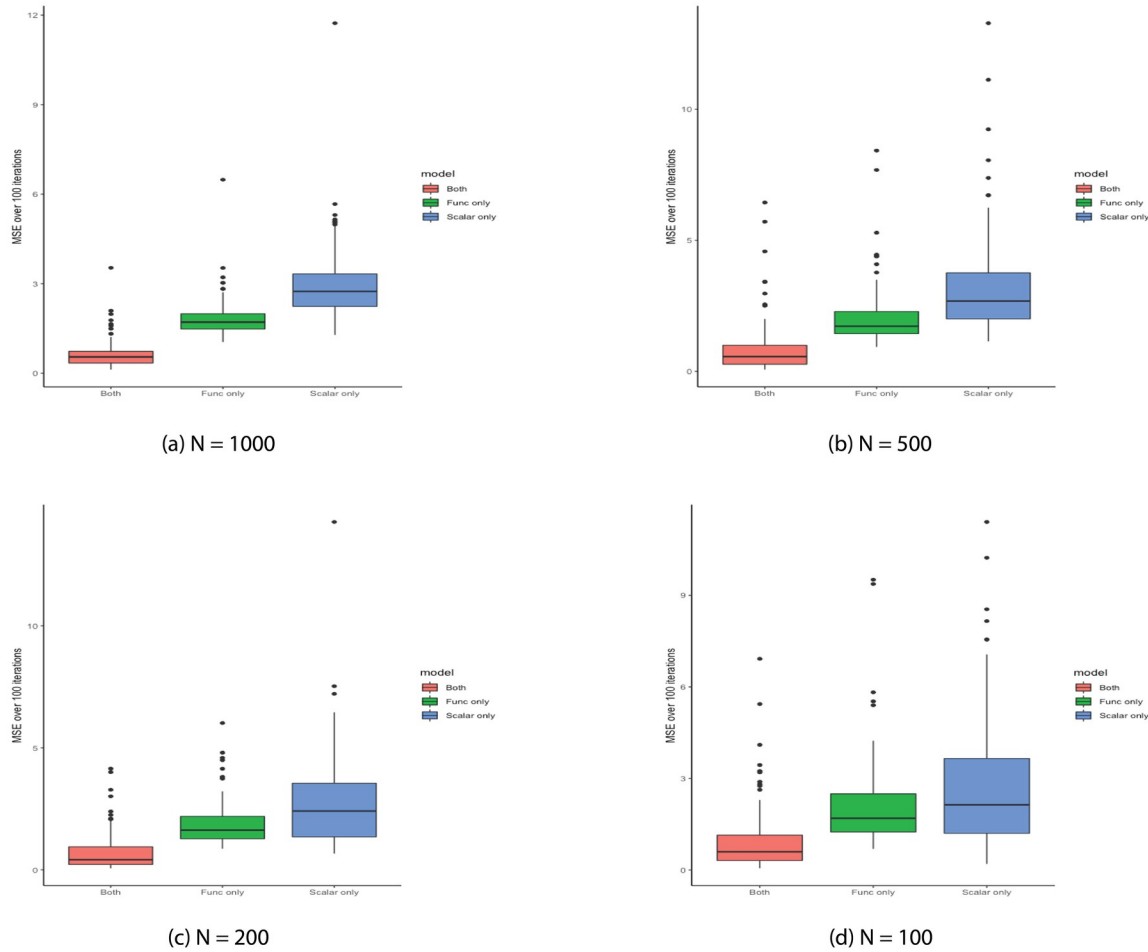

(a) N = 1000

(b) N = 500

(c) N = 200

(d) N = 100

**Fig 7. Mean squared errors (MSE) across three models.** Distribution of the MSEs for each of the three models: AFCM model with both scalar and functional terms (red), AFCM model with only functional term (green), regular Cox model with only scalar term (blue) at different sample sizes (A) N = 1000, (B) N = 500, (C) N = 200, (D) N = 100. The lower the MSE, the better the predictive performance.

ovarian cancer, and TNBC. In particular, we can flexibly use mark connection functions (mcf) as a means to capture the level of stromal infiltration in the lung cancer dataset or immune infiltration in the TNBC dataset with respect to cell distance. In addition to the mcfs, continuous marker expression across cell types (e.g., tumor, immune, stromal, etc.) can be calculated using Moran's I statistics. Thus, qualitative and/or quantitative spatial summary functions can be included in the AFCM, depending on the research question of interest. Furthermore, the advantage of integrating the spatial structure of cells in the TME and patient-level clinical information is demonstrated in the simulation study at different sample sizes.

Though the additive model framework provides us flexibility to the model nonlinear relationship between spatial cell distribution and overall survival, there exist a few limitations. Such complex model requires a relatively large number of parameters to be estimated, which compromises our power to detect significant results provided our limited sample size, as we discuss in Sections 3.1 and 3.2. Recall that the study by Cui et al. [31] demonstrating the application of AFCM in quantifying the association between physical activity and survival data included 2816 participants. Additionally, as with other nonlinear models, the interpretation of results might not always be easy for all applications. We try to draw the connection between our conclusions and Johnson et al.'s that increased immune—tumor cell interactions beyond 100 $\mu$m reduces risk of mortality (Section 3.1). For inter-cell distances in the span of 25–75 $\mu$m radius, on the other hand, our results indicate a different implication that the more tumor cells cluster, the lower the hazard of death. Note that we have specifically chosen mark connection functions (mcf) and Moran's I correlation as representative metrics to summarize qualitative and quantitative spatial information, respectively, in this paper. Other summary functions could be employed to capture various levels of spatial information, depending on the applications. For instance, with the interest in the level of immune infiltration, Barua et al. [29] utilized the G-cross function to quantify the distribution of the nearest immune cells relative to tumor cells within any given inter-cell distance. In such instance, the observed G-cross function could then be used in place of the mcf as functional covariates in the proposed approach.

Motivated by our available datasets, we specifically focus on studying the TME and how spatial heterogeneity of cells within the TME impacts patient overall survival in this article. However, the proposed framework could certainly be extended to other analyses in two directions. First, the spatial architecture of cells being investigated is not necessarily restricted to the TME. Second, other types of response variables other than survival outcomes could be modelled, with necessary modifications made to the model parameterization and estimation procedure indeed. We take the study of human endocrine pancreas and immune system in type 1 diabetes (T1D) using images of pancreatic tissue sections collected from imaging mass cytometry (IMC) platform by Wang et al. [44] as an example. With abundance and localization of proteins at single-cell resolution available, the spatial interaction of immune and pancreatic epithelial cells in pancreatic islets could be captured by the mark connection functions (mcf). Furthermore, with the donors in the dataset being categorized into "controls" (i.e., no history of diabetes) and "T1D" groups, one could use a logistic regression model with the resulting mcfs between immune and pancreatic epithelial cells served as functional covariates, to make inferences on how the islet spatial architecture differs between the two groups and to gain insights into the progression of T1D. By switching to such outcomes from the exponential family, our proposed model would essentially become a functional generalized additive model, which McLean et al. [36] and Muller et al. [45] have discussed in detail.

We have presented promising results associating image summary spatial statistics with subjects' survival, however, our approach relies on the isotropic assumption, which only takes into account the relative distance between any two cells while disregarding their relative coordinations in a given image. Additionally, due to limited sample size, interactions between markers

have not yet been considered in the model. Further modifications are needed to overcome such limitations. Based on locations and associated marker expression intensities of all the cells within an image, multivariate kernel density function may be used to capture the distribution of marker interactions jointly [46]. More precisely, by specifying a search radius bandwidth (e.g., 30 $\mu$m), a joint density representing marker interaction at a given location in an image is estimated using only cells within the radius such that closer cells are weighted more than distant cells. Simultaneously, the number of cells per unit area defined by the bandwidth serves as a multiplier of the cell density values. Such density functions are used as a means to reflect spatially varying marker interactions across images. Due to the nature of density functions, which are non-negative and integrate to one, they cannot be treated as unconstrained functional predictors as in Section 2.2. Intermediate steps to transform the kernel density functions are necessary. A recent approach introduced by Petersen et al. [47] could be used to map regular density functions into a new space through functional transformations including log quantile density and log hazard transformations. Finally, the transformed density functions are incorporated in the model as functional covariates in addition to clinical predictors as before to investigate the association between spatial heterogeneity in marker interactions and subject outcomes. This work is currently under investigation.

After this paper had been submitted, we discovered a study by Wilson et al. [48], which discussed issues with imaging data collected from tissue microarrays (TMA). In particular, the authors highlighted the plausibility of the tissue slides being folded or torn during the slicing process, leading to some sections of the image with no cells present(e.g., panel (C) of S4 Fig). To overcome such challenge, Wilson et al. [48] introduced a framework (with the accompanying R package SpatialTIME [49]) that utilized the K-function in a permutation-based setting to account for such drawbacks. Inspired by the study, it would be interesting to extend our proposed framework to include the permutation-based summary function. More specifically, we would randomly permutate the cell labels to obtain the empirical distribution of the mcfs. Then, the mean mcf instead of the observed counterpart would then be used as functional covariate in the additive functional Cox model (AFCM) to investigate the association between spatial heterogeneity of cells and patient overall survival.

Similar to other spatial analyses, our proposed approach relies on the cell-segmented data. In other words, if cells are not segmented correctly from raw images, the resulting spatial distribution of cells might not accurately reflect the underlying tissue architecture. Accordingly, results of any downstream analyses associating spatial summary information with a clinical outcome of interest would certainly be impacted. As we briefly discuss in Section 2.3, Johnson et al. [35] utilized the state-of-the-art commercial software, inForm, for cell and nucleus segmentation for the NSCLC dataset, while Keren et al. [39] adapted a convolutional neural network approach, DeepCell [38], to extract segmented data in the TNBC dataset. While it is possible that errors could potentially occur with either segmentation approach, this has not thoroughly been investigated and is beyond the scope of our current analysis.

## Supporting information

**S1 Text. A complete analysis for the ovarian cancer dataset.**
(PDF)

**S1 Fig. MHCII expression in NSCLC.** Distribution of MHCII expression in tumor (turquoise) vs. stromal (red) cells in (A) A representative MHCII$^{hi}$ sample. (B) A representative MHCII$^{lo}$ sample.
(TIF)

**S2 Fig. Representative TNBC tissue samples.** Top Row: Pixel-level images of (A) Patient 1 and (B) Patient 2; color-coded from cell segmentation process to be associated with cell-level data. Bottom Row: Corresponding cell-level images for (C) Patient 1 and (D) Patient 2. Each color represents a cell classification group. Dot size is proportional to cell size.
(TIF)

**S3 Fig. P53 expression in TNBC.** Distribution of P53 expression in tumor (turquoise) vs. immune cells (red) in: (A) A representative P53 positive sample. (B) A representative P53 negative sample. (C) Corresponding Moran's I correlation using P53 expression in P53 positive (red) and P53 negative (black) samples. Moran's I values above 0 indicate a direct relationship in P53 expression between tumor and immune cells. Negative Moran's I values suggest an inverse association in P53 expression between tumor and immune cells.
(TIF)

**S4 Fig. Representative images in the ovarian cancer dataset.** Example images of four representative patients (A) Patient 1, (B) Patient 51, (C) Patient 75, and (D) Patient 103, with red and turquoise points denoting stromal and tumor cells, respectively.
(TIF)

**S5 Fig. Ovarian cancer dataset.** (A) Mcf curves for all patients. Note that mcf values below 1 indicate strong clustering of cells of same type, while values above 1 suggest higher level of mixing in of the two cell types. (B) Moran's I correlation between tumor and stromal cells across subjects using Ki67 marker expression. Moran's I values above 0 indicate a direct relationship in Ki67 expression between tumor and stromal cells. Negative Moran's I values suggest an inverse association in Ki67 expression between tumor and stromal cells.
(TIF)

**S6 Fig. Estimated surfaces from AFCM using ovarian cancer dataset.** (A) Estimated surface from AFCM using mcf curves as functional covariates, with values of $\hat{F}$ decreasing from positive (red) to negative (blue). (B) Estimated surface from AFCM using Moran's I correlation in Ki67 expression between tumor and stromal cells, with values of $\hat{F}$ decreasing from positive (red) to negative (blue). Positive $\hat{F}$ corresponds to increased risk of mortality while negative $\hat{F}$ associates with reduced hazard of death.
(TIF)

## Acknowledgments

We thank the Human Immune Monitoring Shared Resource and support of the University of Colorado Human Immunology and Immunotherapy Initiative for their expert assistance in multiplex IHC and generation of the ovarian and lung datasets.

## Author Contributions

**Conceptualization:** Thao Vu, Julia Wrobel, Debashis Ghosh.

**Data curation:** Benjamin G. Bitler, Erin L. Schenk, Kimberly R. Jordan.

**Formal analysis:** Thao Vu, Julia Wrobel, Debashis Ghosh.

**Funding acquisition:** Benjamin G. Bitler, Erin L. Schenk, Debashis Ghosh.

**Investigation:** Thao Vu, Julia Wrobel, Debashis Ghosh.

**Methodology:** Thao Vu, Julia Wrobel, Debashis Ghosh.

**Project administration:** Debashis Ghosh.

**Resources:** Julia Wrobel, Benjamin G. Bitler, Erin L. Schenk, Kimberly R. Jordan, Debashis Ghosh.

**Software:** Thao Vu.

**Supervision:** Julia Wrobel, Debashis Ghosh.

**Validation:** Thao Vu, Julia Wrobel, Debashis Ghosh.

**Visualization:** Thao Vu.

**Writing – original draft:** Thao Vu, Julia Wrobel, Debashis Ghosh.

**Writing – review & editing:** Thao Vu, Julia Wrobel, Benjamin G. Bitler, Erin L. Schenk, Kimberly R. Jordan, Debashis Ghosh.

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
