## [Decision Letter · Decision Letter 0]

2 Dec 2021

Dear Dr. Vu,

Thank you very much for submitting your manuscript "SPF: A Spatial and Functional Data Analytic Approach to cell Imaging data" for consideration at PLOS Computational Biology.

As with all papers reviewed by the journal, your manuscript was reviewed by members of the editorial board and by several independent reviewers. In light of the reviews (below this email), we would like to invite the resubmission of a significantly-revised version that takes into account the reviewers' comments.

We cannot make any decision about publication until we have seen the revised manuscript and your response to the reviewers' comments. Your revised manuscript is also likely to be sent to reviewers for further evaluation.

Sincerely,

Martin Meier-Schellersheim

Associate Editor

PLOS Computational Biology

Jason Haugh

Deputy Editor

PLOS Computational Biology

Reviewer's Responses to Questions

**Comments to the Authors:**

Reviewer #1: The authors have proposed a novel approach to modelling the spatial interactions between cells in the tumour microenvironment (TME) to identify patterns associated with patient survival. The effectiveness of this approach has been demonstrated on publicly available data. While the focus of the manuscript is on the TME, this approach is clearly generalisable to other contexts and, because of this, I will likely be integrating components of their analysis strategy into any future analysis I perform.

## “Estimates from AFCM plots”

-- Could you center all of the colours on zero? Ie, red positive and blue negative.

## Line 122. “Specifically, tumor cells and their immediate neighboring stromal cells are negatively correlated in MHCII expression when they are less than 10 microns apart. As the cross-type cell distance increases, the correlation rises at different rates, and eventually drops close to 0.”

-- Given the average diameter of a cell, what does MHC11 becoming negatively correlated when cells are less than 10 microns mean? Are there worries about poor segmentation?

-- It would be helpful if the figure legends were more verbose, informative and/or repeated some of the information in the text. For example “Fig 4. Estimated surface from AFCM using lung cancer dataset. (a) Mcf curves and (b) Moran's I correlation using MHCII expression”. It would be very helpful for a reader (me) if you fleshed out how to interpret each axis, even though it is mentioned in the text. For example, what does a large mcf or small mcf mean? What does a positive (red) or negative (blue) F() mean?

## Line 181 “In particular, if tumor cells clustered together in the neighborhood of 25 { 75_m, the overall 182

survival improved, based on the negative estimate of the log hazard ratio. However, at distances beyond 183

75 _m, the more tumor cells clustered together, the higher the risk of mortality, while the increased 184

infiltration level of tumor-associated stromal cells reduced the hazard of death.”

-- A few points below:

1) This conclusion would be easier to make if the F() in Figure 4 was centered on white.

2) I am not a fan of the terminology “clustered together”. My preference would be something like, “if strong clustering of tumor cells is observed in the neighborhood of 25 - 75_m”. Or clustering more than expected maybe?

3) The difference in F() appears much larger for r>100 than 25<r<75. be="" considered="" interpreting="" is="" of="" or="" plots="" should="" significance="" some="" sort="" that="" there="" these="" threshold="" viewing="" when="">4) Can you elaborate any more on the complete flipping of interpretation/association that happens around r = 90 in 4a and 4b? It is a hard concept to wrap my head around.

5) As 4a and 4b are coloured by F(), would it also be useful for interpretation for 3b to be moved to Figure 4? Can the lines in Figure 3b also be coloured? Does that make any sense at all? Perhaps you could leave 3b in Figure 3, but then include it in Figure 4 with the top 10% “best survival” patients highlighted in red and bottom 10% “worst survival” patients highlighted in blue? Or some other highlighting that makes sense.

## Figure 5.

-- Why are 5c and 5d two plots? Wouldn’t it make more sense for them to be one plot coloured by mixed/not mixed?

## On line 246 “The authors concluded that P53 positivity associated with negative prognostic significance in breast cancer patients.” “In particular, a sample was defined as P53 positive if any cancer cells was positively expressed. “

-- Pan et al's conclusions appear to be about P53 positive cancer cells. Can you explain what it means for P53 to be correlated between cancer and immune cells? Is this a problem with cell segmentation? Or does P53 have the same function in immune cells as it does cancer cells?

-- The introduction could potentially benefit from some edits to make it easier and quicker to digest. For instance, in the paragraph starting on line 14, I would suggest spending more time talking about multiplexed imaging techniques and avoid detouring to mention bulk/sc sequencing without context.

-- This approach clearly extends beyond studying the TME, I’m not sure how or if it is worth emphasizing this?

-- The authors don’t appear to refer to p-values / statistical significance at any point. Are the “Estimates from AFCM plots” purely for qualitative interpretation? If I were to use your method for my own analysis, is there a single value I can report to provide evidence of a relationship? How would someone use these results?

-- The authors make reference to multiplex data which can quantify many cell types simultaneously however their examples only compare two types of cells (cancer vs immune). The authors briefly mention expanding the tests to include multiple or correlated objects in the conclusion. However, have the authors considered how they would simply apply this approach to multiple pairwise cell type comparisons? Or testing many markers for these comparisons? Is there a sensible way to rank pairwise comparisons? Or would I need to look at all pairwise plots and make a qualitative judgement?

-- Overall, I found the manuscript easy to read. However, I will leave it to the editors to comment on the structure as it does not follow the “manuscript organisation” outlined on PLOS Computational Biology’s website. Some components of the text could be moved from the results into the methods section and the discussion (conclusion) section.</r<75.>

Reviewer #2: This study aim to illustrate the use of established spatial statistics in studying cellular interactions. They motivate their work by discussing the role of tumour microenvironment in cancer evolution. They explore several spatial statistics in different datasets.

In general, the concepts presented in the paper are interesting. However, the presented results show that these metrics are not really very useful in reflecting the biology. For example: “if tumor cells clustered together in the neighborhood of 25 – 75μm, the overall 182 survival improved, based on the negative estimate of the log hazard ratio. However, at distances beyond 183 75 μm, the more tumor cells clustered together, the higher the risk of mortality”. This does not make sense as increased cancer cell density should reflect a worse outcome. Similarly, other statements regard the other analyses is again difficult to interpret. This applies to other statements in the paper.

Major concerns:

- except for the introduction, the paper is very hard to follow. It reads as disparate pieces of experiments that are put together and no clear link between the sections.

- Showing few survival values independent of any other information (e.g Fig. 3C) does not add any information.

- It is not clear how surface plots can be useful.

- It was not clear why simulation was used at the end.

- It was not clear why only the centres of the cells were shown and no original cell images.

**Have the authors made all data and (if applicable) computational code underlying the findings in their manuscript fully available?**

Reviewer #1: Yes

Reviewer #2: **No: **In my experience data is available upon request = “not available” . Code is available

PLOS authors have the option to publish the peer review history of their article (what does this mean?). If published, this will include your full peer review and any attached files.

Reviewer #1: **Yes: **Ellis Patrick

Reviewer #2: No
---

## [Decision Letter · Decision Letter 1]

25 Apr 2022

Dear Dr. Vu,

Thank you very much for submitting your manuscript "SPF: A Spatial and Functional Data Analytic Approach to cell Imaging data" for consideration at PLOS Computational Biology. As with all papers reviewed by the journal, your manuscript was reviewed by members of the editorial board and by several independent reviewers. The reviewers appreciated the attention to an important topic. Based on the reviews, we are likely to accept this manuscript for publication, providing that you modify the manuscript according to the review recommendations.

Sincerely,

Martin Meier-Schellersheim

Associate Editor

PLOS Computational Biology

Jason Haugh

Deputy Editor

PLOS Computational Biology

[LINK]

Reviewer's Responses to Questions

**Comments to the Authors:**

Reviewer #1: Thank you again for the opportunity to review this manuscript. The authors have addressed all of my comments and in doing so I believe the manuscript is more accessible.

Reviewer #2: The authors addressed most of the reviewers’ comments which improved the manuscript.

Comments:

- The proposed metrics can reveal interesting insights, however these are not always easy to interpret. It will be useful to discuss the limitations in the discussion.

- The authors did not report on the methods they used for image analysis which only appeared in their response to reviewer 1. Multiplexed images can be extremely challenging and reasonable segmentation accuracy is needed for reliable conclusions. Details of the image analysis and results (raw images and segmented images) should be included. I appreciate the paper is presenting a method for studying the resulting spatial single cell distribution. However, making it clear in the discussion of potential errors and challenges in data segmentation should be taken into consideration when interpreting the results.

- Following from that point, the red data points that in Figure 1 for Patient 131 are expected to be wrongly segmented cells as TMAs are usually cut as a round biopsy. These should be excluded from the data through QC. Could the author confirm this from the raw images or replace with another example.

- That AFCM does not result in any statistically significant signatures when using this metric, raises a concern regarding their utility of these metrics which make the study merely exploratory. Could the authors elaborate on the potential reason in the discussion?

- Fig. 2:

The text is difficult to read

The authors refer to r=10 in the text, it will be helpful to add a vertical line to indicate this on the figure

- There are few mistakes in referring to the supplementary material: e.g. Line 185 should refer to Fig S2 not S1 and line 283 should refer to Fig. S3 not S1. Line 298 should refer to Section 3 not 2.

- I am not sure why the ovarian dataset and the associated analysis was moved to the supplementary. It should be fit in the appropriate sections of the study.

**Have the authors made all data and (if applicable) computational code underlying the findings in their manuscript fully available?**

Reviewer #1: Yes

Reviewer #2: Yes

PLOS authors have the option to publish the peer review history of their article (what does this mean?). If published, this will include your full peer review and any attached files.

Reviewer #1: **Yes: **Ellis Patrick

Reviewer #2: No

Figure Files:

Data Requirements:

Reproducibility:

References:

---

## [Decision Letter · Decision Letter 2]

16 May 2022

Dear Dr. Vu,

We are pleased to inform you that your manuscript 'SPF: A Spatial and Functional Data Analytic Approach to cell Imaging data' has been provisionally accepted for publication in PLOS Computational Biology.

Best regards,

Martin Meier-Schellersheim

Associate Editor

PLOS Computational Biology

Jason Haugh

Deputy Editor

PLOS Computational Biology

Reviewer's Responses to Questions

**Comments to the Authors:**

Reviewer #2: The authors addressed most of our comments.

**Have the authors made all data and (if applicable) computational code underlying the findings in their manuscript fully available?**

Reviewer #2: Yes

PLOS authors have the option to publish the peer review history of their article (what does this mean?). If published, this will include your full peer review and any attached files.

Reviewer #2: No

---

## [Editor Report · Acceptance letter]

9 Jun 2022

PCOMPBIOL-D-21-01681R2 

SPF: A Spatial and Functional Data Analytic Approach to cell Imaging data

Dear Dr Ghosh,

I am pleased to inform you that your manuscript has been formally accepted for publication in PLOS Computational Biology. Your manuscript is now with our production department and you will be notified of the publication date in due course.

With kind regards,

Zsofia Freund
